# On the Optimal Calculation of the Rice Coding Parameter

**Fernando Solano Donado** [1,2]

[1]  Faculty of Electronics and Information Technology, Warsaw University of Technology,
    00-665 Warsaw, Poland; fs@tele.pw.edu.pl or fs@bluetechnologies.pl; Tel.: +48-222-34-7636
[2]  Blue Technologies, 02-684 Warszawa, Poland

**Abstract:** In this article, we design and evaluate several algorithms for the computation of the optimal Rice coding parameter. We conjecture that the optimal Rice coding parameter can be bounded and verify this conjecture through numerical experiments using real data. We also describe algorithms that partition the input sequence of data into sub-sequences, such that if each sub-sequence is coded with a different Rice parameter, the overall code length is minimised. An algorithm for finding the optimal partitioning solution for Rice codes is proposed, as well as fast heuristics, based on the understanding of the problem trade-offs.

**Keywords:** data compression; Internet of Things; low-power data compression; Micromole; Rice encoding; wireless sensor networks

---

## 1. Introduction

In this article, we study the calculation of the optimal Rice coding parameter. A Rice code is a parameterised Variable-Length Code (VLC)—proposed by Rice in [1,2]—that works in the following way.

Rice encoding: For a given parameter $r$ and source data $n_i$, the following two values are calculated:

$$q_i = \left\lfloor \frac{n_i}{2^r} \right\rfloor, \text{ and} \tag{1a}$$

$$m_i = n_i \mod 2^r, \text{ where} \tag{1b}$$

$\lfloor x \rfloor$ is the floor function: the function that takes as input a real number and returns as output the greatest integer less than or equal to $x$. Similarly, we will use $\lceil x \rceil$ for the ceiling function: the function that takes as input a real number and returns as output the smallest integer higher than or equal to $x$. The function $a \mod b$ refers to the modulo operation: the remainder after the integer division of $a$ by $b$.

The Rice encoding for $n_i$ can be constructed in three steps:

1.  append a bit set to one to the output stream, if $n_i$ is a positive number; or a bit set to zero, otherwise,
2.  encode the value of $q_i$ in unary code and append it to the first bit, and finally,
3.  encode the value of $m_i$ as an unsigned integer using only $r$ bits.

The unary code of a value $x$ is a sequence of $x$ bits, all set to the value one plus one additional bit—a delimiter bit—set to zero.

The reader shall note that the encoding operation can be efficiently implemented in CPUs using bit shift and bit masking operations solely, without the need of floating-point operations.

Rice decoding: The decoder must know the Rice coding parameter $r$ that was used for encoding. The following method is used to obtain the values of $q_i$ and $m_i$ from the encoded stream.

1. read the first bit, and determine the sign of the encoded value: one for a negative sign, zero for a positive sign,
2. count the number of consecutive bits set to one (let $q_i$ be that number),
3. discard the delimiter bit (a bit set to zero), and
4. let the following $r$ bits be the value of $m_i$.

Once the values of $q_i$ and $m_i$ are known, the original value $n_i$ is calculated as:

$$n_i = m_i + q_i \cdot 2^r. \qquad (2)$$

The reader shall note that the decoding operation can be efficiently implemented in CPUs using bit shift and addition operations solely, without the need of floating-point operations.

## 1.1. How the Rice Parameter Affects the Compression Ratio

In Wireless Sensor Networks, transmitted frames do not encode a single measurement value. Instead, in order to minimise the overhead caused by headers of telecommunication protocols, a sequence of measurements is stored, coded, and finally transmitted by the device in a single frame. We initially assume in this article that such a sequence is encoded using a single Rice parameter.

The data sequence presented in Table 1 is used to illustrate the effects of choosing the Rice parameter, $r$, on the total amount of bits encoded. Figure 1 shows the the total amount of bits encoded for this sequence for different values of the Rice parameter. It can be observed that the total encoding length equals 76 bits when $r = 2$. If $r = 3$, the total encoding length decreases to 64 bits. If $r = 4$, the total encoding length increases again to 65 bits. Therefore, the optimal parameter $r$ for this dataset is $r^* = 3$.

**Table 1.** Example input dataset ($n$) with $N = 10$ elements. $L(n_i)$ represents the minimum number of bits needed to encode $n_i$, i.e., $L(n_i) = \lfloor \log_2 n_i \rfloor + 1$ using standard binary representation (also known as beta codes).

| $i$ | 1 | 2 | 3 | 4 | 5 | 6 | 7 | 8 | 9 | 10 |
|---|---|---|---|---|---|---|---|---|---|---|
| $n_i$ | 5 | 7 | 4 | 4 | 12 | 15 | 11 | 45 | 54 | 1 |
| $L(n_i)$ | 3 | 3 | 3 | 3 | 4 | 4 | 4 | 6 | 6 | 1 |

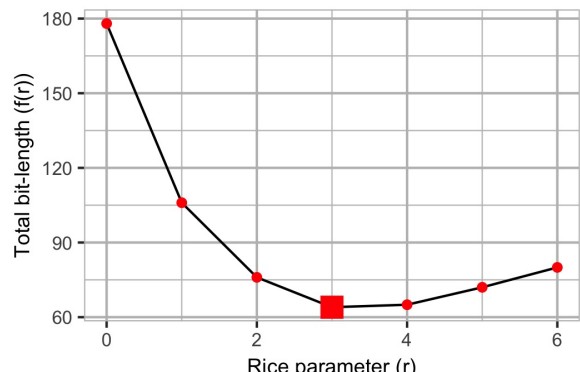

**Figure 1.** Number of bits used to encode randomly generated data by using different Rice parameters $r \in [0, 6]$ for the input data sequence presented in Table 1. The total bit-length function $f(r)$ is only defined for integer values of $r$ (marked with dots and squares in the figure). A continuous line was added between the points to improve visibility.

Table 1 will be used in further sections to provide examples of the proposed heuristics, algorithms, and bounds throughout this article.

### 1.2. Applications

Rice coding has been employed traditionally for data compression of audio [3], images [4–9], video [10,11], and electrocardiogram [12] data. In most cases, the input data is first transformed (using, i.e., a wavelet transform), then the transform coefficients are quantised. The quantised transform coefficients usually follow a two-sided geometric distribution [13,14], for which Rice coding can provide short codes close in length to Huffman codes [15].

In recent years, our research group has developed prototypes of battery-operated resource-constrained Internet of Things (IoT) devices. Such IoT devices usually communicate over low-power lossy wireless networks [16]. In current IoT telecommunication technologies—such as LoRa [17–19]—data rates do not exceed more than 56 kilobits per second, there is no guarantee of successful packet delivery, and the available space for encoding application data in a data frame is no more than 220 bytes [20]. Such small IoT devices may have up to 32 KB of RAM memory with CPUs running at frequencies below 100 MHz [21–23].

In our most recent application, we developed IoT devices for measuring the pH and conductivity of sewage waste water [24,25], with the aim of providing information about the presence and localisation of illegal spills of toxic chemicals in sewers. The pH and conductivity values of aggregated sewage waste water are mostly constant in normal conditions. Based on this, we employed Rice codes for coding the difference between consecutive pH and conductivity measurements of sewage wastewater without employing complex CPU operations or using large amounts of memory.

Several other compression methods have been proposed for resource-constrained IoT devices. Most of the proposed data compression methods in the literature for IoT applications—e.g., [26–31]—are lossy. Those works propose an estimator method that extracts enough information from a time-sequence and then encodes the minimum set of parameters to recreate, as close as possible, the original information based on the transmitted parameters of the estimator method. Ying et al. [32] presented a summary of other compression algorithms for IoT prior to the year 2010.

The focus of this work is to evaluate different methods for calculating the Rice parameter. A comparison study of the compression ratio achieved by other different methods is out of the scope of this article, since it would require evaluating the overhead imposed on the network for transmitting frequency tables, volatile memory usage at constrained nodes, and energy used for coding and decoding.

### 1.3. Contributions

The main contributions of this article are the following:

1. A closed-form expression for the estimation of the optimal Rice parameter in $O(N)$ time, where $N$ is the number of input elements to encode. In contrast to previous works, this estimation applies regardless of the probability distribution of the input data (see Section 3.2);
2. An algorithm for the calculation of the optimal Rice parameter in $O(L_{max} \cdot N)$ time (see Section 3.4), where $L_{max}$ is the minimum number of bits needed for natural coding of the largest integer;
3. An algorithm for the partitioning of the $N$ input elements into sub-sequences yielding a better compression ratio in $O(L_{max} \cdot N^2)$ (see Section 5);
4. Two heuristics for the fast estimation for the partitioning of the input sequence in $O(L_{max} \cdot N)$ (see Section 5.4).

In order to evaluate the performance of the algorithms, the proposed algorithms are further verified and supported with collected data by a real sensor network in Section 4 for the first two contributions in the list above, and again later in Section 6 for the last two contributions in the list above.

## 2. Related Work

In this section, we summarise to the best of our knowledge the research studies carried out so far on Rice coding.

### 2.1. Basic Terminology

As far as the terminology in the area of data compression is concerned, we follow the definitions given by Salomon and Motta in [33].

The terms "source data", "raw data", or "sample" refer to original data obtained from a (sensor) system, before their compression, but after its transformation or quantisation. Throughout this article, we use the term "sequence" to denote a list of consecutive samples captured over time. A "sub-sequence" is a list of consecutive samples found within a sequence. In this article, a sequence of samples is represented with $n$ and the $i$-th element of the this sequence by $n_i$.

The output of a compression algorithm is termed encoded data. Accordingly, a compressor algorithm converts raw data or samples into compressed encoded data of a smaller bit-length.

### 2.2. Parameter Calculation for Data Following Laplacian Distributions

Yeh et al. in [15] analysed the optimality of Rice codes to Huffman codes for Laplacian distributions. Following this study, Robinson in [3] proposed that the optimal Rice parameter for data following a Laplace distribution $r_{[LAP]}$ be given by:

$$r_{[LAP]} = \log_2 \left( \ln(2) E(|x|) \right), \text{ where} \tag{3}$$

$|x|$ represents the absolute value of $x$, $E(x)$ is the expected value of $x$, $\ln(x)$ is the natural logarithm function and $\log_2(x)$ is the binary logarithm function. The value of $E(x)$ is the weighted sum taken over all possible symbols $x$, viz., $\sum |x| \cdot p(x)$, where the weight $p(x)$ corresponds to the probability of occurrence of the symbol $x$ in a sequence.

Merhav et al. extended the study to Two-Sided Geometric Distributions (TSGD) in [13].

### 2.3. Derivation for Data Following Geometric Distributions

Rice coding is a special case of Golomb coding as proposed by Golomb in [34]. It has been shown in [33] that if the set of numeric data that needs to be encoded is assumed to follow a geometric distribution, i.e., the probability of the number $n_i$ occurring in the input sequence is:

$$P(n_i) = p^{n_i} \cdot (1 - p), \tag{4}$$

for some probability $p \in (0, 1)$, then the optimal Golomb parameter is given by the integer number $m_G$ that minimises the following expression:

$$p^{m_G} - 1/2. \tag{5}$$

Therefore, the value of $m_G$ can be calculated as:

$$m_G = \left\lceil -\frac{\log_2 (1 + p)}{\log_2 p} \right\rceil. \tag{6}$$

As Rice encoding is the special case of Golomb encoding when the parameter $m_G$ of the Golomb coder is a power of two, i.e., $m_G = 2^r$, such an inequality is resolved for Rice encoding to:

$$r_{[GEO]} = \log_2 \left\lceil -\frac{\log_2 (1 + p)}{\log_2 p} \right\rceil. \tag{7}$$

Under the same assumptions, viz. of following a geometric probability distribution, Kiely in [35] proposed the selection of the code parameter considering only the mean value of the distribution as follows:

$$k_{geo}^* = \max\left\{0, 1 + \left\lfloor \log_2\left(\frac{\ln(\phi - 1)}{\ln(\frac{\mu}{\mu+1})}\right)\right\rfloor\right\}, \text{where} \tag{8}$$

$\mu$ corresponds to the estimate of the mean value of the data and $\phi$ is the golden ratio constant value $(\sqrt{5}+1)/2$.

Previously, other authors considered the case when the input data follow a geometrical distribution and provided a means of calculating the optimal Golomb encoding parameter, hence providing a bound for the Rice encoding parameter. Clearly, these bounds do not apply for all distributions.

In this work, we provide bounds on the value of the Rice encoding parameter regardless of the type of distribution of the data.

### 2.4. Adaptive Rice Coding

Malvar in [6] and later improved in [7] proposed the Run-Length/Golomb–Rice (RLGR) coder. RLGR was designed to work best when the input sequence follows a TSGD. RLGR uses both run-length and Rice coding for encoding an input sequence of measurements. Run-length coding is used for compressing sequences of measurements with a value of zero.

The Rice coding parameter is automatically adapted after a sample is coded: if the encoding process using the previous parameter yields zero on the value of $q_i$, the Rice coding parameter is increased for coding the next symbol; if the previous parameter yields a value higher than one on $q_i$, the Rice coding parameter is decreased proportionally. In our opinion, this idea works best if the consecutive values to be encoded are close to each other, but may yield sub-optimal Rice codes in the case that the consecutive values to be encoded are considerably different.

In Section 5, we propose a way for efficiently using Rice codes in situations where the input sequence of values changes abruptly.

## 3. General Selection of the Golomb Parameter for Rice Coding

We now focus on the problem of how to find the value of the Rice parameter $r$ that yields the minimum bit-length of the output stream for any raw data sequence $\boldsymbol{n}$. The variation in the bit-length due to the choice of this parameter was illustrated above with an example in Section 1.1.

To simplify notation, we assume that we deal only with the Rice encoding of values with their sign. In such a case, to apply Rice coding to a number $n_i$, a total of $r + \left\lfloor \frac{n_i}{2^r} \right\rfloor + 2$ bits are required. Hereinafter, since the sign is encoded in a separate bit, we assume that the samples in the input sequence $\boldsymbol{n}$ have been already stripped off its sign; they are all non-negative.

In this section, we are interested in finding the value of $r$ for which the value of the following function:

$$\begin{aligned}
f(r, \boldsymbol{n}) &= \sum_{1 \le i \le N}\left(r + \left\lfloor \frac{n_i}{2^r}\right\rfloor + 2\right) \\
&= (r+2)\cdot N + \underbrace{\sum_{1 \le i \le N}\left\lfloor \frac{n_i}{2^r}\right\rfloor}_{Q(r)},
\end{aligned} \tag{9}$$

is minimum. Note that $f(r, \boldsymbol{n})$ is a discontinuous function since its values are only valid for integer values of $r$. In this section, an analytical solution is provided to this problem.

Let us define the function $L(n_i)$ as the number of bits used for representing $n_i$:

$$L(n_i) = \begin{cases} \lfloor \log_2 n_i \rfloor + 1, & \text{if } n_i \neq 0 \\ 0, & \text{otherwise.} \end{cases} \tag{10}$$

The values of $L(n_i)$ can be observed for our example sequence in Table 1.

It is clear that the optimal value for $r$, $r^*$, for $f(r, \boldsymbol{n})$ is an integer value bounded by:

$$\min_i L(n_i) \leq r^* \leq \max_i L(n_i). \tag{11}$$

Hereinafter, we name $L_{\min}$ and $L_{\max}$ the left- and right-hand sides expressions of (11), respectively.

Moreover, note that for each positive number to encode, $n_i$, the integer division expression can be bounded as follows:

$$\frac{n_i}{2^r} - \phi_i \leq \left\lfloor \frac{n_i}{2^r} \right\rfloor \leq \frac{n_i}{2^r} - \theta_i, \tag{12}$$

for certain values of $\phi_i$ and $\theta_i$.

We create two derivable continuous functions that will bound the value of $f(r, \boldsymbol{n})$ as follows:

$$g(r, \boldsymbol{n}) \leq f(r, \boldsymbol{n}) \leq h(r, \boldsymbol{n}), \text{ where} \tag{13}$$

$$\begin{aligned} g(r, \boldsymbol{n}) &= (r+2) \cdot N + \sum \left( \frac{n_i}{2^r} - \phi_i \right) \\ &= (r+2) \cdot N + \frac{\sum n_i}{2^r} - \phi, \end{aligned} \tag{14}$$

and:

$$\begin{aligned} h(r, \boldsymbol{n}) &= (r+2) \cdot N + \sum \left( \frac{n_i}{2^r} - \theta_i \right) \\ &= (r+2) \cdot N + \frac{\sum n_i}{2^r} - \theta, \end{aligned} \tag{15}$$

for certain values of $\phi$ and $\theta$.

To simplify notation, we drop the parameter $\boldsymbol{n}$ (the input data sequence) from the functions $f, g,$ and $h$ since it remains constant in what remains of our analysis.

It is not difficult to prove that $g(r)$ and $h(r)$ are both convex functions in $r$. As an example, we illustrate in Figure 2 the two bounding functions for the Rice parameter for the example given in Section 1.1, together with the optimal values of each function (indicated using a larger squared region).

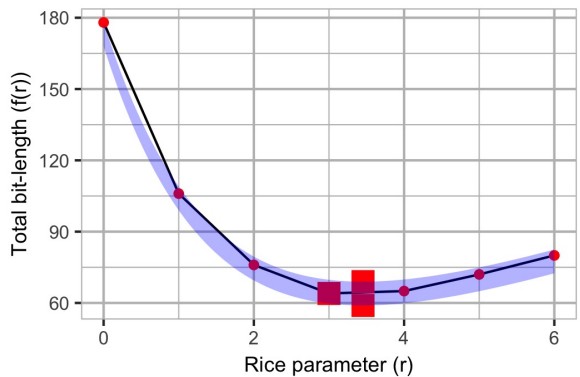

**Figure 2.** Function $f(r)$ bounded by $g(r)$ and $h(r)$ for the input sequence of Section 1.1. The minimum value of each function is marked with a squared red dot. The total bit-length function $f(r)$ is only defined for integer values of $r$. A continuous line was added between the points to improve visibility.

Therefore, in order to find the solution, the evaluation of the functions for consecutive values of *r* until the output value starts increasing is not a very time-consuming task. Nonetheless, in further subsections, we will present a faster method for finding the optimal Rice parameter.

### 3.1. Function Bounds

Since $g(r)$ is a derivable function, its first derivative is:

$$\frac{\partial g(r)}{\partial r} = N - \ln 2 \cdot \frac{\sum n_i}{2^r}. \tag{16}$$

Since $g(r)$ is convex, we can find the value of $r_g^*$ yielding the minimum value of the function by setting the function's first derivative to zero:

$$N = \ln 2 \cdot \frac{\sum n_i}{2^{r_g^*}}, \tag{17}$$

which means that:

$$r_g^* = \log_2 \left( \frac{\ln 2}{N} \sum n_i \right). \tag{18}$$

Since $h(r)$ is a derivable function, we can proceed with the same approach again to find its optimal value. In fact, its first derivative is the same as the first derivative of $g(r)$, which means that:

$$r_h^* = \log_2 \left( \frac{\ln 2}{N} \sum n_i \right). \tag{19}$$

For nomenclature convenience, let us define $S(\boldsymbol{n})$ as the right-hand side expression of (18) and (19):

$$S(\boldsymbol{n}) = \log_2 \left( \ln(2) \mu \right), \text{ where} \tag{20}$$

$\mu$ is the arithmetic mean of the values in $\boldsymbol{n}$. Please note the similarities of (20) with (3), and the different assumptions concerning the probability distributions of the input data for deriving these two expressions.

### 3.2. Conjecture and Approximation

Since $\ln 2$ is an irrational number, the quantity $S(\boldsymbol{n})$ is also irrational.

We conjecture that the optimal integer parameter for the Rice code is bounded by:

$$\lfloor S(\boldsymbol{n}) \rfloor \leq r^* \leq \lceil S(\boldsymbol{n}) \rceil + 1 \text{, if } S \geq 0$$
$$r^* = 0 \text{, otherwise.} \tag{21}$$

In our validation results, in Section 4, we will evaluate how closely the conjectured expression provides the optimal parameter for data compression of real datasets.

### 3.3. Pre-Calculation of Q

Any non-negative integer value $a$ is represented in binary by a sequence of bits $\boldsymbol{c} = \{c_j\}$ such that:

$$a = \sum_{0 \leq j \leq L(a)} c_j \cdot 2^j, \ c_j \in \{0, 1\}. \tag{22}$$

Let us denote by $q(x, r)$ the function performing the integer division of $x$ by a power of two—viz. right bit shift. Such an operation can be defined as follows in terms of (22):

$$q(a, r) = \left\lfloor \frac{a}{2^r} \right\rfloor = \sum_{j \geq r} c_j \cdot 2^{j-r}. \tag{23}$$

It is not difficult to notice that our definition fulfils the following recursive property of bit shifting:

$$q(a,r) = \begin{cases} q(q(a,r-1),1), & \text{for } 1 \leq r < L(a) \\ a, & \text{for } r = 0. \end{cases} \qquad (24)$$

Considering the encoding problem of this article, we can redefine $Q(r)$ as follows:

$$Q(r) = \sum_i \left\lfloor \frac{n_i}{2^r} \right\rfloor = \sum_i q(n_i, r). \qquad (25)$$

We are interested in designing an algorithm that efficiently updates the value of $Q(r)$ as new samples are added to the set $\boldsymbol{n}$. Let us denote by $\boldsymbol{n'}$ the set of samples $\boldsymbol{n}$ enlarged by a new sample $\hat{n}$. The value of $Q(r)$ for the new set is therefore defined as:

$$Q(r, \boldsymbol{n'}) = Q(r, \boldsymbol{n}) + q(\hat{n}, r). \qquad (26)$$

The next section presents an algorithm that is defined based on (24) and on (26).

### 3.4. Algorithm for Finding the Optimal Parameter

In Algorithm 1, we propose a search heuristic for finding the optimal Rice parameter starting from our approximation values.

It starts by comparing the bit-lengths of the Rice coding of the input sequence when the parameters are $\lfloor S(\boldsymbol{n}) \rfloor$ and $\lceil S(\boldsymbol{n}) \rceil$. If the total bit-length resulting from Rice coding using the first parameter ($\lfloor S(\boldsymbol{n}) \rfloor$) is smaller than when the second value ($\lceil S(\boldsymbol{n}) \rceil$) is used, the search continues using values smaller than $\lfloor S(\boldsymbol{n}) \rfloor$ until the bit-length starts increasing again. If the total bit-length caused by the first parameter is greater than when the second value is used, the search continues using values greater than $\lceil S(\boldsymbol{n}) \rceil$ until the bit-length starts increasing again. If the total bit-lengths are the same, an optimum has been reached.

The function FindBestRiceParameter—defined between Lines 17 and 37 in Algorithm 1—is our main function in the heuristic. It shall be invoked with the input data sequence, $\boldsymbol{n}$, and the number of elements in the input data sequence, $N$.

The subroutine PrecalculateRiceQ—defined between Lines 1 and 12 in Algorithm 1—has a complexity of $O(L_{max} \cdot N)$, since each one of the $n_i \in \boldsymbol{n}$ input data values is considered, and for each one of them, the value of $Q(r)$ is calculated for all potential values of $r$, i.e., $0 \leq r \leq L_{max}$. The operator $a >> b$ is the right-shift-bit operator or, in other words, the integer division by a power of two. The subroutine CalculateF—defined between Lines 13 and 16 in Algorithm 1—has a complexity of $O(1)$, since it is a mathematical evaluation with no iterative calls on the input data. The loop defined between Lines 30 and 35—within the subroutine FindBestRiceParameter—has only a complexity of $O(L_{max})$, since the subroutine is evaluating the function $f$ with all the possibles values of $r$ in constant time. As a consequence, the complexity of the subroutine FindBestRiceParameter is defined by the invocation of the subroutine PrecalculateRiceQ in Line 18, which has a higher complexity than the aforementioned loop. The subroutine FindBestRiceParameter has a complexity of $O(L_{max} \cdot N)$.

---

**Algorithm 1** Online calculation of the Rice-code with complexity $O(L_{max} \cdot N)$.

---

 1: **function** PRECALCULATERICEQ($n$)
 2:     $Q \leftarrow 0$
 3:     **for** $n_i \in n$ **do**
 4:         $r \leftarrow 0$
 5:         **while** $r \leq L(n_i)$ **do**                                    ▷ see (10) for $L(x)$
 6:             $Q(r) \leftarrow Q(r) + n_i$
 7:             $n_i \leftarrow n_i >> 1$
 8:             $r \leftarrow r + 1$
 9:         **end while**
10:     **end for**
11:     **return** $Q$
12: **end function**

13: **function** CALCULATEF($r, Q, N$)
14:     $f \leftarrow N \cdot (r + 2) + Q(r)$
15:     **return** $f$
16: **end function**

17: **function** FINDBESTRICEPARAMETER($n, N$)
18:     $Q \leftarrow$ PRECALCULATERICEQ($n$)
19:     $S \leftarrow \log_2 \left( (\ln 2) \cdot Q(0)/N \right)$
20:     $r' \leftarrow \lfloor S \rfloor$
21:     $f' \leftarrow$ CALCULATEF($r', Q, N$)
22:     $r'' \leftarrow \lceil S \rceil$
23:     $f'' \leftarrow$ CALCULATEF($r'', Q, N$)
24:     $t \leftarrow +1$
25:     **if** $f' < f''$ **then**
26:         $t \leftarrow -1$
27:         SWAPVARIABLES($r', r''$)
28:         SWAPVARIABLES($f', f''$)
29:     **end if**
30:     **while** $f' > f''$ **do**
31:         $r' \leftarrow r''$
32:         $f' \leftarrow f''$
33:         $r'' \leftarrow r'' + t$
34:         $f'' \leftarrow$ CALCULATEF($r'', Q, N$)
35:     **end while**
36:     **return** $r'$
37: **end function**

---

In practice, as will be presented later in Section 4.4, the approximation described in Section 3.2 (used as a starting search point in our algorithm) provides the optimal solution. In our experimental results, the loop between Lines 30 and 35 of the algorithm was executed only once or never at all, since either of the values of $r'$ and $r''$ was yielding the optimal value of $r$ before the first loop iteration.

Taking advantage of the fixed maximum number of bits that a microcontroller can use for representing an integer number, the author will work on a hardware implementation of the function PrecalculateRiceQ, which will allow decreasing the complexity of the whole procedure to $O(N)$ by using $L_{max}$ hardware counters.

## 4. Performance on Different Datasets Using Single-Parameter Rice Coding

In this section, a numerical validation of the heuristics and approximations from Section 3 is presented.

### 4.1. Datasets and Preprocessing

The datasets used for our numerical analysis are the sets of measurements from the weather stations located at beaches along Chicago's Lake Michigan lakefront. The weather stations have measured air temperature, wet bulb temperature, humidity, rain intensity, interval rain, total rain,

precipitation type, wind direction, wind speed, maximum wind speed, barometric pressure, solar radiation, heading, and the station battery life from three different locations once per hour since the 22nd of June 2015. By considering different sensor types and the different station locations, there are in total 37 datasets. Due to the occasional failure of some sensors or stations, the number of measurements in each dataset ranges from 27,058 to 35,774.

We believe that the way data are preprocessed has an affect on the compression factor of the Rice coding. In this article, we consider three simple methods for data preprocessing.

In the first preprocessing method—referred to as "normalize_dataset_scale"—all measurements are scaled by a constant factor so as to avoid having fractional values. This is needed since Rice coding does not have the possibility to encode non-integer (fractional) values. The scaling factor was calculated for each dataset as the minimum difference between any pair of measurements. All values are then divided by this scaling factor.

In the second preprocessing method—referred to as "normalize_dataset_scale_diff"—all measurements are scaled as previously mentioned, and then, the difference between consecutive scaled measurements is taken. Since we expect that some observed phenomena will change slowly over time, the author believes that the optimal Rice parameter for the difference of consecutive scaled measurements can be smaller, yielding shorter codes on average. This reflects the estimator methods used in some IoT data compression algorithms, such as in [36].

In the third preprocessing method—referred to as "normalize_dataset_shift_mean"—all measurements are scaled as mentioned in the first preprocessing model, and then, the difference between the scaled measurements and the mean of all scaled values in the batch is taken. This reflects the estimator methods used in some IoT data compression algorithms, such as in [36].

### 4.2. Compression Factor of Analysed Codes

As a starting point, the effect of the preprocessing method used before encoding is analysed for Rice coding. Here, we consider only the optimal Rice coding solution.

In Figure 3 is shown the average length needed for encoding a single measurement for each dataset and considering the three preprocessing functions mentioned above. The usage of the second preprocessing function provides shorter codes in all cases, with the exception of the interval rain dataset.

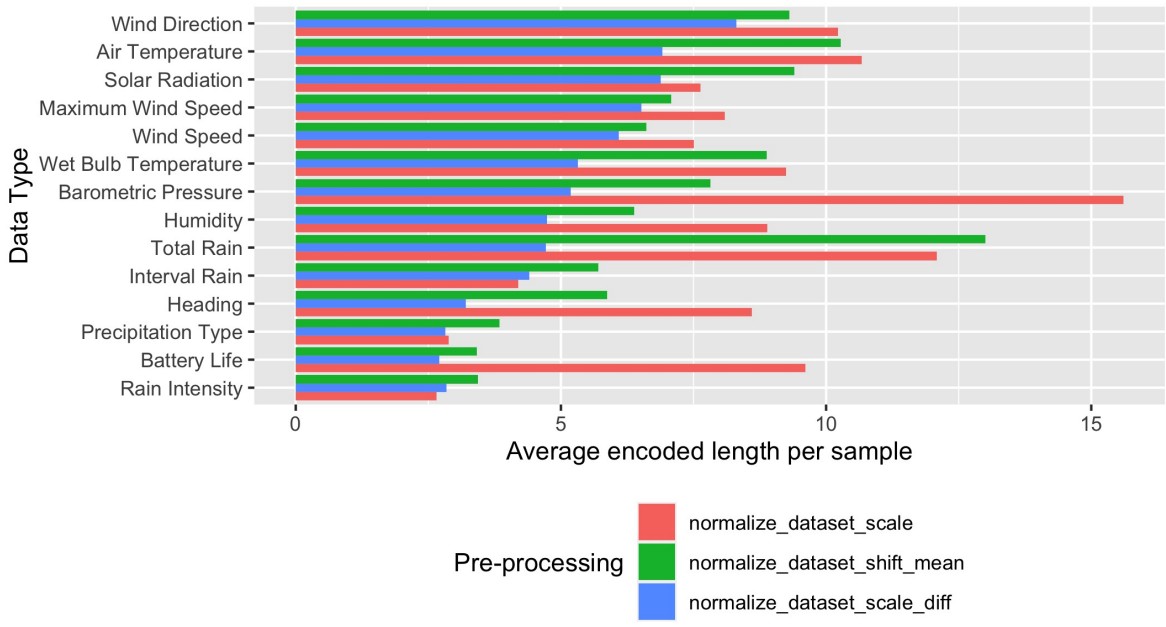

**Figure 3.** Estimation of the minimum bit-length after Rice coding.

### 4.3. Estimation of the Output Bit-Length

We conjecture that an estimation of the minimum output bit-length after Rice coding can be derived from (9), by substituting the value of the Rice parameter given by the value provided in (20), as follows:

$$\hat{f}(\boldsymbol{\mu}, \boldsymbol{\sigma}) = N \cdot \left[ \log_2(\ln(2) \cdot \mu) + 2 + \left| \frac{\sigma}{2^{\ln(2) \cdot \mu}} \right| \right], \text{ where} \tag{27}$$

$\mu$ and $\sigma$ are the mean and standard deviation of the samples to be encoded.

Figure 4 shows the output bit-length found after using the algorithm proposed in this article for finding the minimum output length and the estimated output bit-length based on (27). The size of the points is proportional to the ratio of the output bit-length based on Shannon theorem and the optimal output bit-length after Rice coding.

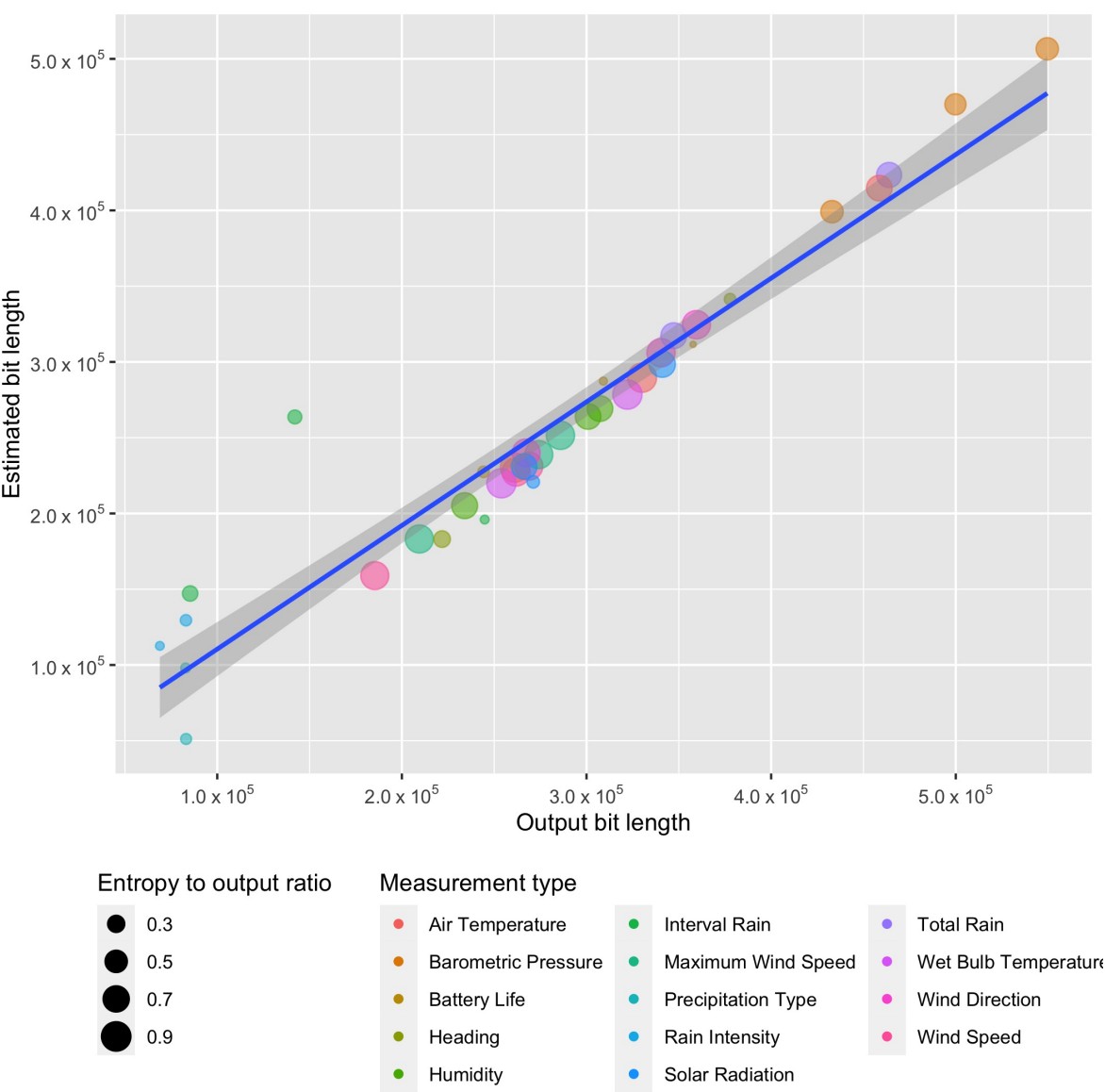

**Figure 4.** Estimation of the minimum bit-length after Rice coding when the first preprocessing method is used.

A linear approximation was found with the equation $f = 1.126 \cdot \hat{f} - 9781.446$, with a standard deviation error of 0.056 and a *t*-value of 19.91 for the estimated slope value, for the input sequences after being preprocessed with the first preprocessing method.

### 4.4. Approximation Solutions

In this subsection, we illustrate whether the conjecture in (20) is valid.

Equation (20) provides three different potential integer values for the optimal Rice coding parameter: $\lfloor S \rfloor$, $\lceil S \rceil$, and $\lceil S \rceil + 1$. In Figure 5 is shown the count of solutions found by approximating the Rice coding parameter as mentioned in (20).

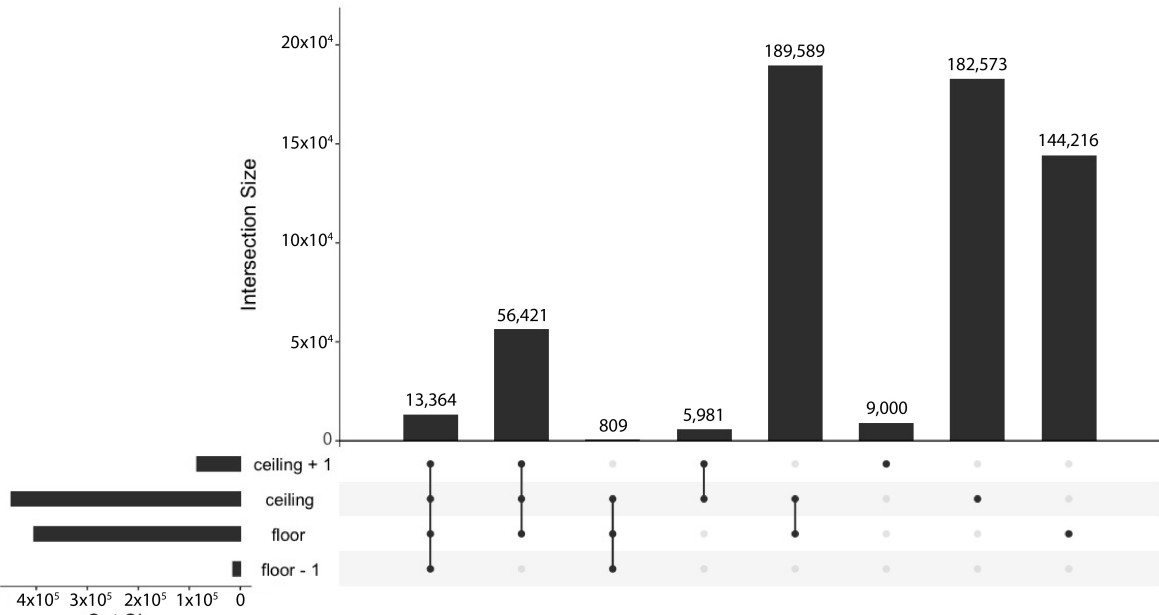

**Figure 5.** Number of minimal bit-length solutions found by approximating the Rice parameter as given in (20). Labels "floor", "ceiling", and "ceiling + 1" should be understood as $\lfloor S \rfloor$, $\lceil S \rceil$, and $\lceil S \rceil + 1$, respectively. Each column represents the number of minimal bit-length solutions obtained by using the specified approximations, e.g., there were 144,216 minimal bit-length solutions found by approximating the parameter as $\lfloor S \rfloor$ and 189,589 minimal bit-length solutions found by approximating the parameter either as $\lfloor S \rfloor$ or $\lceil S \rceil$ (both are equal-cost solutions).

All minimal bit-length solutions were found by approximating the Rice coding parameter: the optimal Rice coding parameter was found using $\lfloor S \rfloor$ in 64.8%, $\lceil S \rceil$ in 71.2%, and $\lceil S \rceil + 1$ in 11.9% of the 601,953 cases.

### 4.5. Influence of the Chosen Batch Size

The compression factor $\tau_m(\boldsymbol{n})$ of an encoding method $m$ for an input dataset $\boldsymbol{n}$ of length $N$ is defined as follows [33]:

$$\tau_m(\boldsymbol{n}) = \frac{N \cdot \max_i L(n_i)}{f_m(\boldsymbol{n})}, \text{ where} \tag{28}$$

$L(n_i)$ is the number of bits needed to represent the sample $n_i$ in binary code and $f_m(\boldsymbol{n})$ is the output bit-length size of the encoding method $m$ when used over $\boldsymbol{n}$.

In Figure 6, the normalised compression factor for each data type is shown as the batch size varies. The normalised compression factor is calculated as:

$$\hat{\tau}_m(\boldsymbol{n}) = \frac{\tau_m(\boldsymbol{n}) - \min_{|k|=|n|} \tau_m(\boldsymbol{k})}{\max_{|k|=|n|} \tau_m(\boldsymbol{k}) - \min_{|k|=|n|} \tau_m(\boldsymbol{k})} \tag{29}$$

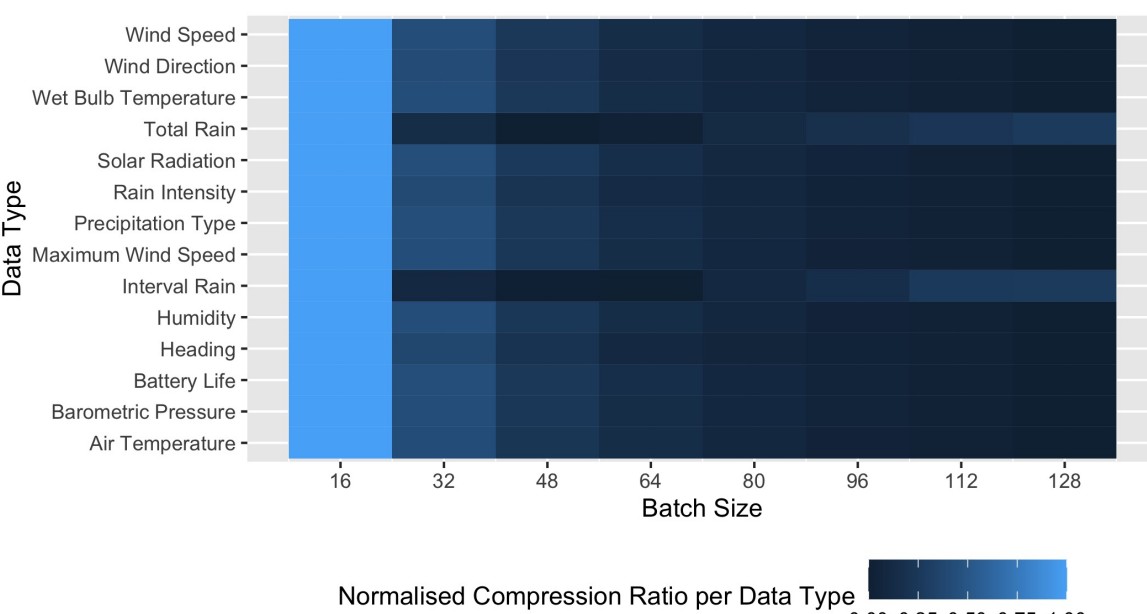

**Figure 6.** Normalised compression factor for each data type as the batch size varies.

As can be observed for most datasets, the compression ratio of Rice coding may decrease for some datasets as the batch size increases, yielding longer codes on average. This is because a single Rice coding parameter cannot fit well to a large sequence of measurements.

For the total rain and interval rain datasets, the compression ratio fluctuates as the batch size varies, indicating that the sequence of measurements largely varies over time.

## 5. Multi-Parameter Rice Coding

As shown in Section 4.5, when a large data sequence comprises samples of different orders of magnitude, Rice coding using a single parameter may be inefficient. This is because there is no Rice parameter that can be set efficiently for the whole sequence: a large Rice parameter may aid in the reduction of the bit-length of samples having large values at the expense of increasing the bit-length of small samples, while a small Rice parameter may aid in the reduction of the bit-length of samples having small values at the expense of increasing the bit-length of large samples.

As an example, we show in Figure 7 a sequence of 128 measurements taken from the interval rain dataset. Most of the values of this sequence are close to zero, but a few peak values are found at the beginning of the sequence. The optimal Rice coding parameter for this particular sequence is $R^* = 4$, yielding a code of 987 bits. Instead, if we are allowed to partition the sequence of 128 values into sub-sequences, in such a way that a different Rice coding parameter can be used for each sub-sequence, it could probably yield a shorter code (after taking into account the overhead of encoding the extra parameters). The optimal solution for this instance of the partitioning problem is represented in the same figure with coloured boxes. It consists of partitioning the sequence into four sub-sequences and Rice coding them with parameters 5, 0, 9, and 0, respectively. This solution can encode the same sequence in 359 bits, after considering an overhead of 16 bits for appending to the stream the corresponding value of the Rice parameter for each sub-sequence.

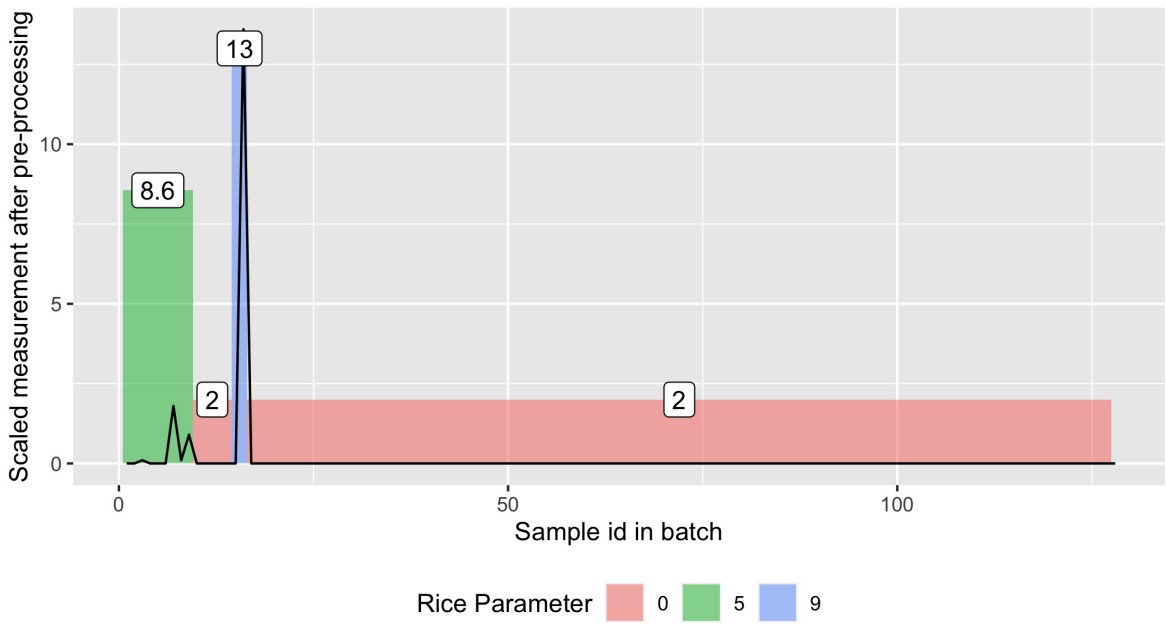

**Figure 7.** Scaled sequential difference of 128 measurements of interval rain from the 63rd Street Weather Station in Chicago, USA, between 09:00:00 and 19:00:00 local time on 1 March 2017. The continuous (black) line represents the sequence of measurements over time. The shaded (coloured) boxes represent the way in which the given sequence of measurements can be partitioned into four sub-sequences in order to achieve shorter Rice codes.

More formally, for a given input sequence of samples, $\boldsymbol{n}$, a more optimal encoding can be achieved if we partition the sequence of input data, $\boldsymbol{n}$, into a set $\mathscr{P} = (\boldsymbol{p}_k)$ of $P$ consecutive sub-sequences and we allow ourselves to encode each sub-sequence $\boldsymbol{p}_k$ with a different Rice parameter $r_k$. We would like the reader to notice that these two decision problems—viz. how each sub-sequence is defined and what Rice parameter is to be used for each sub-sequence—are interrelated: the choice of the ranges defining each sub-sequence boundaries has an effect on the value of its optimal parameter for efficient Rice coding and vice versa.

In this section, we present an algorithm for finding the optimal bounds for each sub-sequence and the corresponding Rice parameters so as to minimise the overall total bit-length.

The algorithm is divided into three phases. In the first phase, in Section 5.1, a set of special sub-sequences is calculated, namely basis. In the second phase (Section 5.2), the algorithm calculates the costs of all feasible sub-sequences in an efficient manner, by reusing previous calculations of other sub-sequences (including the basis). In the third phase (Section 5.3), the algorithm constructs an auxiliary Directed Acyclic Graph (DAG) with at most $2 \cdot N + 2$ weighted nodes. The optimal solution to the problem results from finding the shortest path in the auxiliary weighted DAG.

While this approach is generally tedious, simply because it would require the calculation of the costs of all possible sub-sequences, it will help us in setting up the basic concepts for two efficient heuristics suitable for implementation in constrained IoT devices. These efficient heuristics will be presented later in Section 5.4.

*5.1. Basis—Reducing Input Length*

Initially, we attempt to reduce the input length of our problem. For this, we partition the input sequence of $N$ samples into $\hat{P} \leq N$ consecutive sub-sequences of samples, in such a way that the samples in each sub-sequence require the same number of bits for encoding, as given by the value of the function $L(n_i)$ previously defined in (10).

In this section, the sub-sequences are defined by the pair of indices of the start and end positions in its original input sequence $n$. We illustrate the outcomes of this process with the example dataset previously presented in Table 1. In this case, a total of four sub-sequences—viz. $[1, 4]$, $[5, 7]$, $[8, 9]$ and $[10, 10]$—shall conform the basis, since the values of $L(n_i)$ for the samples within these ranges is the same.

In this first step, for each sub-sequence in the basis, we compute and store: the value of $Q(r)$ of (9) for different values of $r$, the range index in the sequence, and the sum of the values of that sub-sequence. This information will be used to generate all the other potentially needed sub-sequences in the second step of this algorithm (see Section 5.2). The function for computing these parameters can be seen in Algorithm 2. For each sub-sequence $p_k$, the function returns the index of its starting and last elements in $u$ and $v$, respectively. Note that $u_{k+1} = v_k + 1$, since no sample is lost in the partitioning, nor repeated. The function also returns in $q$ the encoding length of the each one of the $\hat{P}$ sub-sequences for different potential values of $r$, and also the sum of the values for each sub-sequence $\hat{s}_k$.

---

**Algorithm 2** Calculation of the basis for the multi-parameter Rice-encoding problem with complexity $O(L_{max} \cdot N)$.

---

1: **function** MULTIPARAMETERBASIS($n$, $N$)
2:　　$k \leftarrow 1$
3:　　$i' \leftarrow 1$
4:　　$l' \leftarrow L(n(1))$　　　　　　　　　　　　　　　　　　　　　　$\triangleright$ see (10) for $L(x)$
5:　　$q \leftarrow (0)$
6:　　$\hat{s} \leftarrow (0)$
7:　　$L_{min} \leftarrow \min_i L(n(i))$
8:　　$L_{max} \leftarrow \max_i L(n(i))$
9:　　**for** $1 \leq i \leq N$ **do**
10:　　　　$\hat{s}_k \leftarrow \hat{s}_k + n(i)$
11:　　　　**for** $L_{min} \leq r \leq L_{max}$ **do**

12:　　　　　　$q_{kr} \leftarrow q_{kr} + \left\lfloor \frac{n(i)}{2^r} \right\rfloor$

13:　　　　**end for**
14:　　　　**if** $L(n(i)) \neq l'$ **then**
15:　　　　　　$u_k \leftarrow i'$
16:　　　　　　$v_k \leftarrow i$
17:　　　　　　$k \leftarrow k + 1$
18:　　　　　　$i' \leftarrow i + 1$
19:　　　　　　$l' \leftarrow L(n(i'))$
20:　　　　**end if**
21:　　**end for**
22:　　**return** $\{q, \hat{s}, u, v, k, L_{min}, L_{max}\}$
23: **end function**

---

Algorithm 2 has two nested loops. The outer loop—between Lines 9 and 21—iterates over all input values. The inner loop—between Lines 11 and 13—calculates the values $q_{kr}$ for every potential value of $r$ for the input value in consideration. Therefore, this step has a complexity of $O(L_{max} \cdot N)$.

The output of Algorithm 2 for the example data presented in Table 1 is provided in Table 2.

In order to better illustrate constant values in the remaining part of this section, we will use $\mathcal{U}_k$ and $\mathcal{V}_k$ to denote the set of indices of $n$ where the sub-sequence $k$ of the basis starts and ends, respectively. These two values correspond to the output values of $u$ and $v$ of Algorithm 2. In addition, the output value of $k$ becomes the constant $\hat{P}$: the length of the basis and the output matrix $q = (Q_k(r))$ become an array of constants $q(k, r)$.

**Table 2.** The generated $\hat{P} = 4$ sub-sequences for the basis for the example input data with $N = 10$ as a result of Algorithm 2. The values of $Q_k(r)$ represent the value of $Q(r)$ as given in (9) for the $k$-th sub-sequence of the basis, defined by the indices $u_k$ and $v_k$. The column $\hat{s}_k$ provides the sum of the values of the input data within the indices $u_k$ and $v_k$.

| $k$ | $Q_k(r)$ | | | | | | $u_k$ | $v_k$ | $\hat{s}_k$ |
|---|---|---|---|---|---|---|---|---|---|
| | $r = 1$ | 2 | 3 | 4 | 5 | 6 | | | |
| 1 | 9 | 4 | 0 | 0 | 0 | 0 | 1 | 4 | 20 |
| 2 | 18 | 8 | 3 | 0 | 0 | 0 | 5 | 7 | 38 |
| 3 | 49 | 24 | 11 | 5 | 2 | 0 | 8 | 9 | 99 |
| 4 | 0 | 0 | 0 | 0 | 0 | 0 | 10 | 10 | 1 |

### 5.2. Computing the Cost of All Sub-Sequences Efficiently

The sub-sequences calculated in the first step do not embrace all potential sub-sequences, but only the shortest ones. The second phase of the algorithm calculates the minimum bit-length of all other possible sub-sequences of $n$, by reusing the information generated after the basis calculation by Algorithm 2.

Algorithm 3 receives as input: an index of the basis sub-sequences ($k_0$), the values of $q(r)$ of the basis for each potential value of $r$ ($q$), the start and end index of the sub-sequence ($u$ and $v$, respectively), the length of the basis ($\hat{P}$), and the range of the values of $L(n_i)$ ($L_{min}$ and $L_{max}$, respectively).

---

**Algorithm 3** Calculation of the cost for all sub-sequences with complexity $O(L_{max} \cdot \hat{P}^2)$.

---

1: **function** MULTIPARAMETERSUBSEQUENCESFROM($k_0, q, u, v, \hat{P}, L_{min}, L_{max}$)
2:　　$w \leftarrow (\infty)$
3:　　$z \leftarrow (0)$
4:　　$y \leftarrow (0)$
5:　　**for** $k_0 \leq k \leq \hat{P}$ **do**
6:　　　　**for** $L_{min} \leq r \leq L_{max}$ **do**　　　　　　　　　　　　　　　　▷ see (10) for $L(x)$
7:　　　　　　$y_r \leftarrow y_r + q(k, r) + (v(k) - u(k) + 1) \cdot (r + 2)$
8:　　　　　　**if** $w_k > y_r$ **then**
9:　　　　　　　　$w_k \leftarrow y_r$
10:　　　　　　　$z_k \leftarrow r$
11:　　　　　**end if**
12:　　　　**end for**
13:　　**end for**
14:　　**return** $\{w, z\}$
15: **end function**

---

The algorithm returns the minimum bit-length $w$ and the best Rice parameter $z$ for each one of the $K - k_0 + 1$ sub-sequences in $n$ that have as first element $\mathcal{U}_{k_0}$.

Algorithm 3 consists of two nested loops. The outer loop—between Lines 5 and 13—considers the creation of a sub-sequence by merging the sub-sequences from $k_0$ until $k$ in $\hat{P}$. Each time it considers the creation of a sub-sequence, the inner loop—between Lines 6 and 12—calculates the bit-length $y_r$ of encoding such a new sub-sequence with different values of $r$. As a result, Algorithm 3 has a running complexity of $O(L_{max} \cdot \hat{P})$.

Algorithm 3 needs to be called once for each sub-sequence in the basis. Therefore, computing the costs of every possible sub-sequence starting at any index of the input data yields a running time complexity of $O(L_{max} \cdot \hat{P}^2)$.

For the example sequence provided in this article, Algorithm 3 is invoked four times (once per sub-sequence in the basis). The values stored in the variable $y_r$ in the first iteration of each call are shown in Table 3. These values correspond to the bit-length of the sub-sequences in the basis for different values of the Rice parameter.

**Table 3.** Total bit-length for each one of the $\hat{P} = 4$ sub-sequences of the basis (shown in Table 2) for each value of the Rice parameter ($r$). Each row of this table is implicitly calculated by Algorithm 3 in its first iteration and stored in variable $y_r$ for further usage.

| Basis | $r = 1$ | 2 | 3 | 4 | 5 | 6 |
|---|---|---|---|---|---|---|
| $[1,4]$ | 21 | 20 | 20 | 24 | 40 | 48 |
| $[5,7]$ | 27 | 20 | 18 | 18 | 21 | 21 |
| $[8,9]$ | 55 | 32 | 21 | 17 | 16 | 17 |
| $[10,10]$ | 3 | 4 | 5 | 6 | 7 | 8 |

Table 4 shows the input values of Algorithm 3 for the example sequences of this article. Each column represents the minimum bit-length and optimal Rice parameter for each one of the 10 possible sub-sequences for the example problem.

**Table 4.** Output data of Algorithm 3. The first header row ($k_0$) represents the starting index of the sub-sequence. The second header row ($k$) represents the end index of the sub-sequence. Each column provides values of the minimum bit-length for the sub-sequence ($w$) and the corresponding value of the Rice parameter ($z$).

| From ($k_0$) | 1 | | | | 5 | | | 8 | | 10 |
|---|---|---|---|---|---|---|---|---|---|---|
| To ($k$) | 4 | 7 | 9 | 10 | 7 | 9 | 10 | 9 | 10 | 10 |
| $w$ | 20 | 38 | 59 | 64 | 18 | 35 | 41 | 16 | 23 | 3 |
| $z$ | 2 | 3 | 3 | 3 | 3 | 4 | 4 | 5 | 4 | 1 |

### 5.3. Auxiliary DAG and Optimal Solution

In the third phase, we construct an auxiliary Directed Acyclic Bipartite Graph (DABiG) for finding the best subset of sub-sequences that yield the best multi-parameter solution.

The DABiG is constructed as follows:

- for each sub-sequence in the basis $k$, we add two vertices $u_k$ and $v_k$, representing the sub-sequence's start $\mathcal{U}_k$ and the $\mathcal{V}_k$, respectively;
- we add two special vertices $v_\alpha$ and $u_\omega$;
- we add one edge from each vertex $u_{k_1}$ to each vertex $v_{k_2}$, where $k_2 \geq k_1$, with the cost given by $w_{k_2}$ when Algorithm 3 is invoked with $k_1$ as its first parameter;
- we add one edge from each vertex $v_k$ to $u_{k+1}$ with cost $\epsilon$; and finally,
- we add one edge from $v_\alpha$ to $u_1$ with cost $\epsilon$ and another edge from $v_M$ to $u_\omega$ with cost zero.

Therefore, the DABiG consists of $2 \cdot \hat{P} + 2$ vertices and $\frac{\hat{P} \cdot (\hat{P}+1)}{2} + \hat{P} + 2$ directed edges. A graphical representation of the DABiG can be observed in Figure 8.

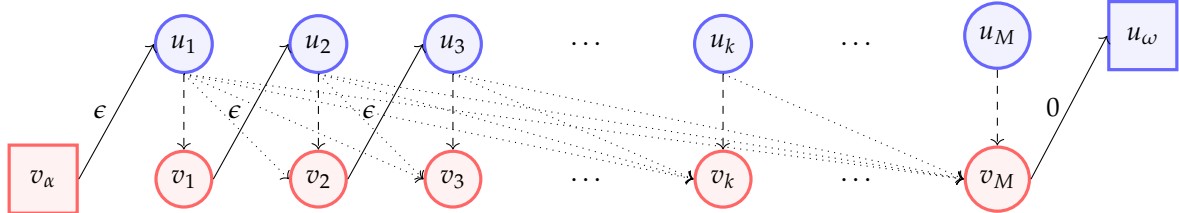

**Figure 8.** The directed acyclic bipartite graph for solving the multi-parameter Rice encoding problem. Each pair of $u_k$ and $v_k$ vertices represents the start and end of a sub-sequence in the basis (see Algorithm 2). The edges represent all possible ways of grouping the input data into larger sub-sequences. Edge costs are equal to the minimum bit-length of the sub-sequence $w_k^{(i)}$ (see Algorithm 3) or to the overhead bit-length caused due to adding an extra sub-sequence to the solution ($\epsilon$). The solution to the problem is given by the shortest path from $v_\alpha$ to $u_\omega$.

The solution to our problem is given by the shortest path from $v_\alpha$ to $u_\omega$. The total minimal cost of the encoding is the cost of the shortest path found.

Following with our example, Figure 9 shows the DABiG based on the found sub-sequences by Algorithm 3, which were previously mentioned in Table 4.

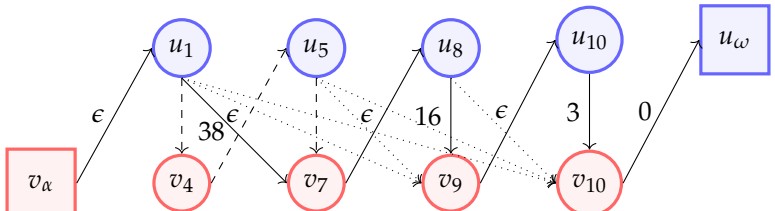

**Figure 9.** The directed acyclic bipartite graph for the previously provided example sequence, with weights ($w$) as given in Table 4. When $\epsilon = 0$, the shortest path has a total cost of 57. This shortest path is marked using continuous lines as arcs between the vertices. Only the weight of the vertices for this shortest path and those equal to $\epsilon$ are shown. When $\epsilon = 8$, the shortest path solution corresponds to the path $v_\alpha \rightarrow u_1 \rightarrow v_{10} \rightarrow u_\omega$ (not shown) with cost $64 + \epsilon = 72$.

### 5.4. Two Simple Heuristics for Constrained IoT Devices

Based on the analysis of the solution to the problem of finding the optimal Rice coding using the partitioning process as described above, in this subsection, we present two parameterized heuristics of reduced complexity in terms of both memory and CPU usage.

The idea behind the design of these parameterized heuristics is to define—in a single iteration through samples—output sub-sequences based on their similarity in terms of the number of bits needed for their encoding, following the concept of the basis presented in Section 5.1. To do this, we assume that we are given an input parameter, say $\Delta$, bounded by $0 \leq \Delta \leq (L_{\max} - L_{\min})$, which we name the spread factor. Output sub-sequences are going to be built in such a way that the difference in the bit-length of the raw values—i.e., $L(n_i)$—of each output sub-sequence will not be larger than the parameter $\Delta$. As a result, if $\Delta$ equals $(L_{\max} - L_{\min})$, the output of these heuristics will be a single sub-sequence comprising all elements. On the other hand, if $\Delta$ is equal to zero, the output of these heuristics will be the set of all the sub-sequences found in the basis (as explained in Section 5.1).

Our first proposed heuristic for this problem sets the encoding parameter of each sub-sequence as the optimal value as calculated by Algorithm 1. Our second proposed heuristic for this problem sets the encoding parameter of each sub-sequence to its respective value of $\lfloor S \rfloor$, as explained in Section 3.2.

The pseudo-code of the second heuristic can be observed in Algorithm 4. The function RiceEncode($n, i, j, r$) does only the encoding (no parameter calculation) of the sub-sequence of $n$ starting from the $i$-th element to the $j$-th element using Rice parameter $r$. Since each element in $n$ is passed through only once and all other calculations are done within the loop itself—between Lines 6 and 20—the running complexity of the second heuristic is $O(n)$.

The pseudo-code for the first heuristic is very similar, with the difference that the estimation $s$ and its associated summation $\hat{s}$ are not implemented and, hence, not passed to the function RiceEncode. In such a case, the function RiceEncode should find the optimal parameter for the given sub-sequence using Algorithm 1, before encoding the set of samples. Each element must be passed twice: first by the loop function presented in Algorithm 4 and later for determining the optimal parameter (Algorithm 1) for the sub-sequence. The running complexity of the first heuristic is, therefore, $O(N + N \cdot L_{\max}) = O(N \cdot L_{\max})$, i.e., it is defined by the complexity of finding the optimal parameter of a sub-sequence.

---

**Algorithm 4** Heuristic for Rice encoding with partitioning with complexity $O(N)$.

---

1: **function** RICEENCODEMULTIHEURS($\boldsymbol{n}$, $N$, $\Delta$)
2:    $i' \leftarrow 1$
3:    $L'_{\max} \leftarrow 0$
4:    $L'_{\min} \leftarrow \infty$
5:    $\hat{s} \leftarrow 0$
6:    **for** $1 \leq i \leq N$ **do**
7:       $l \leftarrow L\big(n(i)\big)$
8:       **if** $(l > L'_{\max} - \Delta)$ and $(l < L'_{\min} + \Delta)$ **then**
9:          $L'_{\max} \leftarrow \max(l, L'_{\max})$
10:        $L'_{\min} \leftarrow \min(l, L'_{\min})$
11:        $\hat{s} \leftarrow \hat{s} + |n|$
12:       **else**
13:          $s \leftarrow \left\lfloor \log_2 \left\{ \frac{\hat{s} \cdot \ln 2}{i - i'} \right\} \right\rfloor$
14:          RICEENCODE($\boldsymbol{n}, i', i - 1, s$)
15:          $i' \leftarrow i$
16:          $L'_{\max} \leftarrow l$
17:          $L'_{\min} \leftarrow l$
18:          $\hat{s} \leftarrow |n|$
19:       **end if**
20:    **end for**
21:    **if** $i' \neq N$ **then**
22:       $s \leftarrow \left\lfloor \log_2 \left\{ \frac{\hat{s} \cdot \ln 2}{N - i' + 1} \right\} \right\rfloor$
23:       RICEENCODE($\boldsymbol{n}, i', N, s$)
24:    **end if**
25: **end function**

---

### 5.5. Multi-Parameter Coding and Decoding

In the previous subsection, we defined different algorithms for the calculation of a set of Rice parameters that can be applied to partitions of the input data sequence. In this subsection, we propose a method for encoding the set of Rice parameters and the generated Rice codes of each sub-sequence within a single bit-stream without ambiguity.

The proposed method encodes each sub-sequence separately by providing first a code for its Rice parameter, then the Rice encoded values in the sub-sequence, and finally, a terminating delimiter for the sub-sequence.

In most applications, the Rice parameter is usually a relatively small value. In such cases, the code for the Rice parameter can be the corresponding Elias gamma code, or any other prefix non-parametrizable VLC code (refer to [33] for a list of some of them). In applications where the Rice parameter is usually large, the author suggests including instead a field of fixed bit-length and using the natural coding for the Rice parameter. In our numerical results in the upcoming Section 6, we will consider the later case and include an overhead of one single byte per sub-sequence for encoding the Rice parameter.

As for the terminating delimiter, it should be recalled that in Section 1, an explanation of Rice coding and decoding for all integer values was given. It should be pointed out that, according to the definition of Rice coding presented here (from [33]), there could be two Rice codes for the input value zero (0), since zero is neither positive nor negative but still Rice coding utilises one bit for coding its sign. In this work, for convention, we propose the usage of negative zero for delimiting the termination of the Rice coding of a sub-sequence and the coding of the Rice parameter of the next sub-sequence. In this way, the delimiter will occupy $r + 2$ additional bits for each sub-sequence. In order to avoid confusion, an input sample with value zero should be always encoded assuming that it is positive.

## 6. Performance on Different Datasets Using Multi-Parameter Rice Coding

In this section, we validate the multi-parameter Rice coding methods presented in previous sections. Our aim is to understand whether there is any benefit from partitioning the dataset for data compression as mentioned earlier. The dataset previously described in Section 4.1 is used in this section.

### 6.1. Cases Where Multi-Parameter Rice Coding Is Beneficial

First, we calculate the optimal partitioning of every sequence previously described, and checked whether such a solution produces shorter codes than the solution when a single Rice coding parameter is used. In nearly 10% of all batches used for validation in this article, a shorter Rice coding could be found by multi-parameter Rice coding.

In Figure 10, we can observe the percentage of the total number of batches for each dataset and batch size for which a shorter code can be found, if sequence partitioning is considered. As expected, the datasets interval rain and rain intensity can substantially be better compressed when a large batch size is considered.

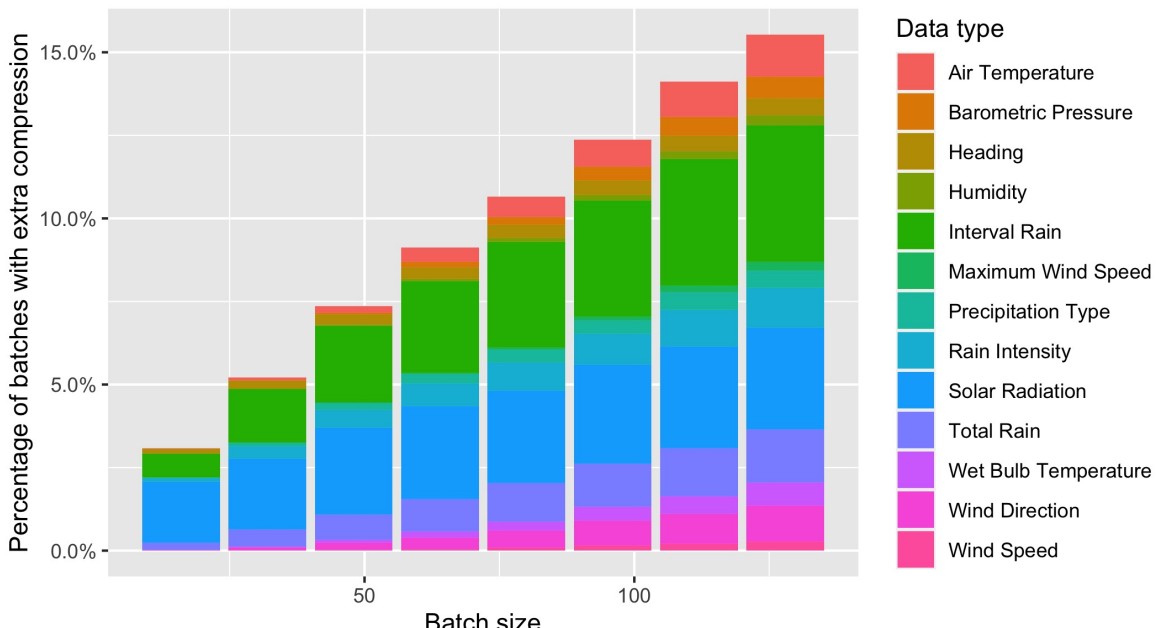

**Figure 10.** Number of cases where multi-parameter Rice coding yields shorter codes than the single-parameter Rice coding approach.

### 6.2. Improved Compression Factor by Multi-Parameter Rice Coding

Next, we analyse how much the compression factor can be improved for those batches where sub-sequence partitioning is meaningful.

In Figure 11, we show the relative difference between the compression factors achieved when the optimal single-parameter and optimal multi-parameter Rice coding approaches are considered. For interval rain and rain intensity, a further compression factor between 20% and 25% can be achieved.

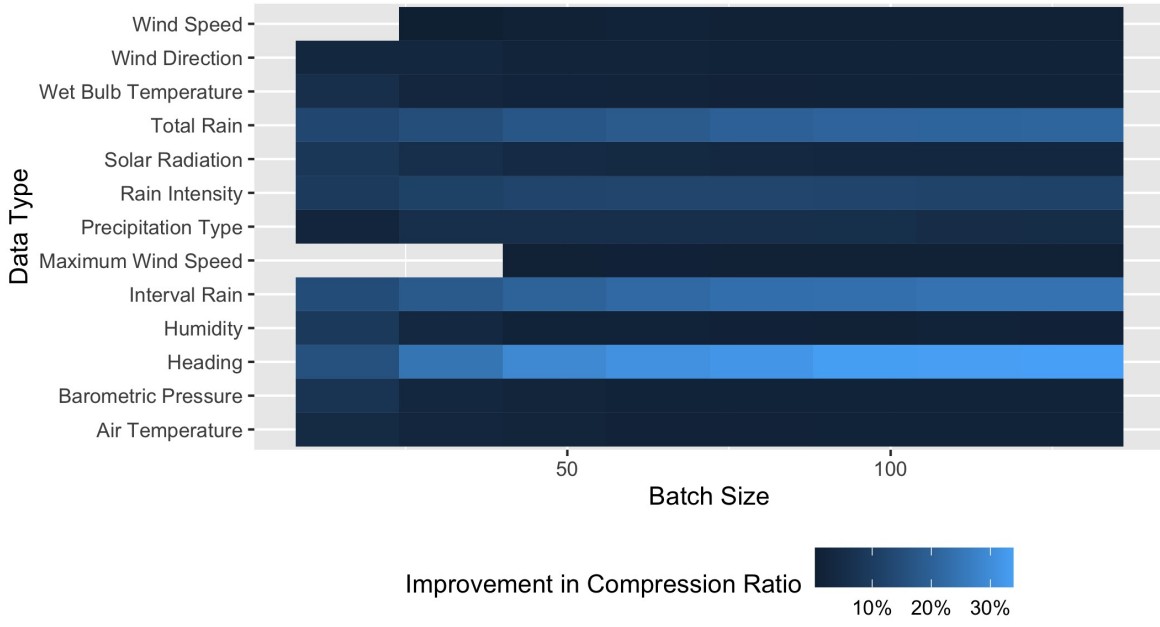

**Figure 11.** Relative improvement of the compression factor when multi-parameter Rice coding is used per data type.

### 6.3. Heuristic Performance

Finally, we illustrate how the second heuristic described in Section 5.4—with lower complexity—is capable of finding the optimal solution to the Rice partitioning partitioning problem.

In Figure 12, the number of batches where the heuristic was able to find the same optimal solution is shown. The maximum number of optimal solutions that are found occurred when $\Delta = 5$. In total, sixty-five percent of the optimal solutions are found when carrying the $\Delta$ parameter from zero to five.

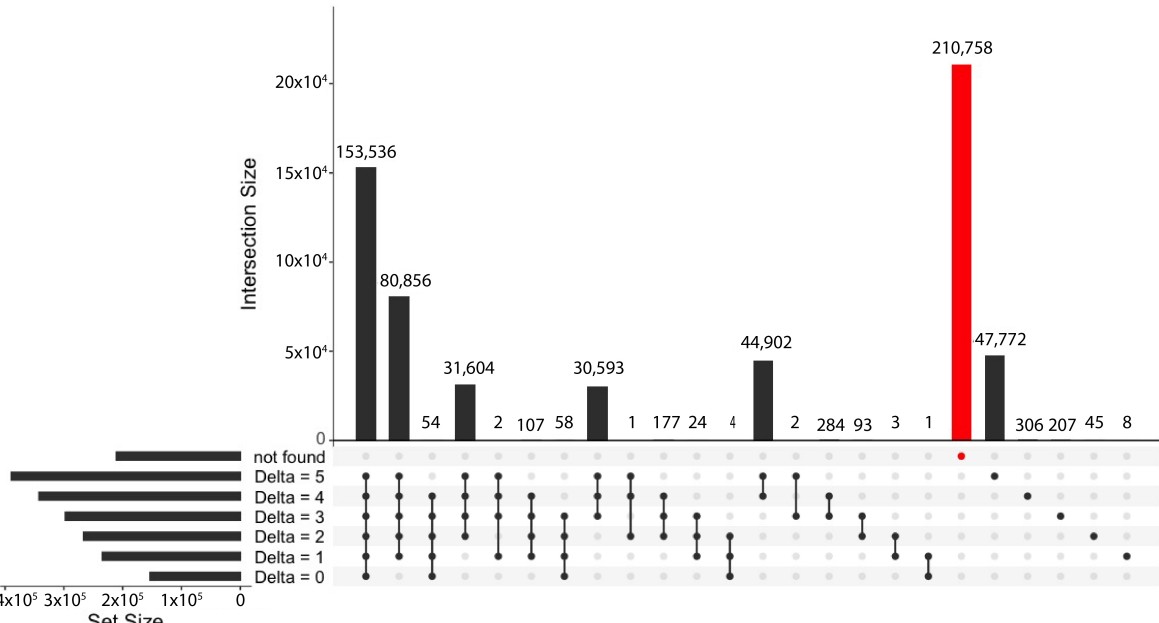

**Figure 12.** Number of minimal bit-length solutions found by the multi-parameter heuristic, as the parameter $\Delta$ varies. For 35% of the tested batches, the heuristic was not able to find the optimal solution.

For the remaining 35% of the solutions, where the optimum partitioning solution could not be found, we analyse how close the heuristic solutions are to its optimal.

In Figure 13, we show a histogram where each cell illustrates the ratio of the optimal solution and the heuristic solution as the batch size varies for different datasets. Empty cells imply that all solutions found by the heuristic were optimal for that batch and dataset combination.

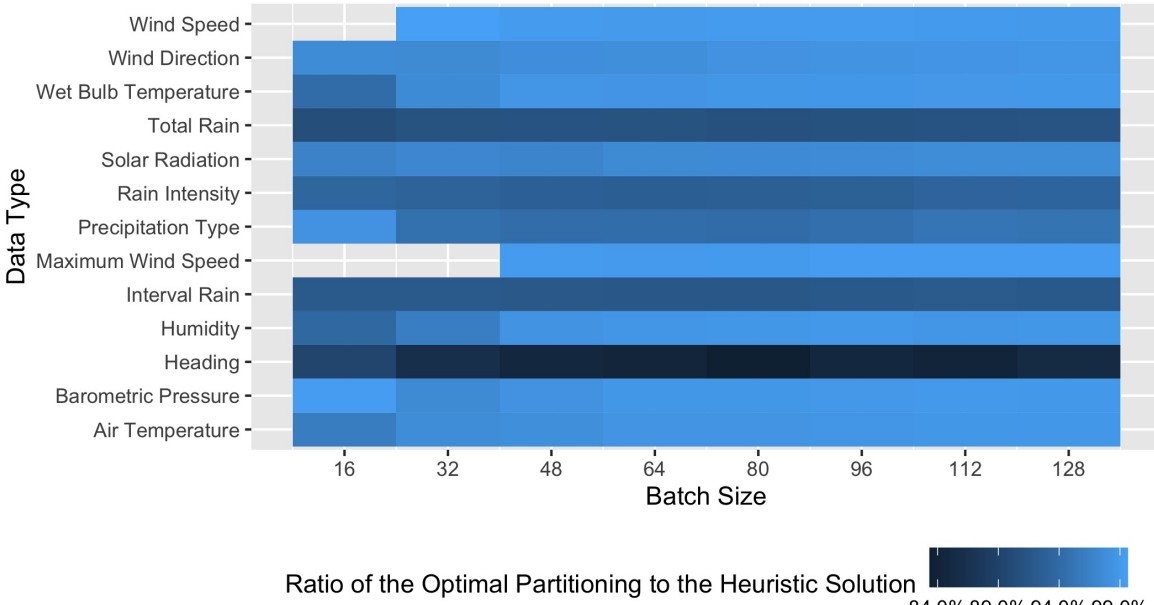

**Figure 13.** Ratio of optimal Rice coding considering partitioning versus the best heuristic partitioning solution found. The mean is calculated considering only those heuristic solutions for which the optimal could not be found, which account for 35% of the batches.

## 7. Conclusions

In this article, we designed several methods for the calculation of the optimal Rice coding parameter and then evaluated them using real data.

Our analysis and numerical verification showed that the optimal Rice coding parameter for a data sequence can be quickly bounded as expressed by (21). In addition, we also observed that the output bit-length of Rice codes can be estimated by (27) knowing the mean and variance of the dataset.

In around 10% of the experiments, it was observed that partitioning the sequence into sub-sequences, such that each sub-sequence is coded with a different Rice parameter, can be profitable. An algorithm for finding the optimal partitioning solution for Rice codes was proposed, as well as fast heuristics, based on the understanding of the problem trade-offs.

**Funding:** This work was supported in part by the FP7 Goldfish project, which received funding from the European Union's Seventh Framework Programme for research, technological development and demonstration under Grant Agreement No. 269985, as well as by the H2020 Micromole [24] and H2020 SYSTEM [25] projects, which have received funding from the European Union's Horizon 2020 research and innovation programme under Grant Agreement No. 653626 and No. 787128, respectively.

**Acknowledgments:** The author would like to thank Javier Delgado Palacios for the implementation of early versions (not presented in this manuscript) of the algorithms described and evaluated in this article.

**Conflicts of Interest:** The author declares no conflict of interest.

## Abbreviations

The following abbreviations are used in this manuscript:

DOAJ    Directory of open access journals
IoT     Internet of Things
kbps    kilobytes per second
KB      Kilobyte
LoRa    Long-Range telecommunication standard
MDPI    Multidisciplinary Digital Publishing Institute
MHz     Megahertz
RLGR    Run-Length/Golomb–Rice
TSGD    Two-Sided Geometric Distribution

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
