# Peer review of "On the Optimal Calculation of the Rice Coding Parameter"

_algorithms, doi:10.3390/a13080181_

Round 1

Reviewer 1 Report

The article ‘On the Optimal Calculation of the Rice Coding’ propose algorithms for the calculation of optimal rice coding parameter. It also validates these algorithms on data coming from multiple sensors, found in the context of real IoT systems. Finally, it also explores the use of multiple rice coding parameter by partitioning the data into subsequences. The article is well written with clear explanations to the different equations and algorithms and adding relevant references to the related works. However, I have some suggestions to the author which I am detailing below.

Though the author has presented the Huffman coding and dictionary methods (Page 3) commonly used for the purpose of data compression as well as their limitations, the suggestion that the Rice coding may be useful for IoT devices (Lines 49-50, 75-77) seems not very convincing to me in the introductory section.

The authors present the algorithmic complexity of the proposed algorithms, without any explanation on how these values were obtained. Take for example, lines 206-209. Even though, for some of the proposed algorithms, it may seem quite evident, it would still be interesting to have a brief description on how these values were obtained.

In the context of using single rice coding parameter, the decoding algorithm is presented (Lines 28-34). However, the decoding in the case of the proposed multi-parameter rice coding is not detailed. How these multiple parameters are handled or transmitted? In case of single rice coding parameter, these may be part of the first few bytes or already by the transmitter and the receiving end. But what happens when you have multiple parameters. There is multiple information like the total number of parameters and the parameter values. It becomes more complicated, especially when we see that sub-sequences may be of varying length (Table 4).  What is the size overhead (in bytes) to store/transmit this information? Are these considered in the evaluation of the algorithms/heuristics?

Some clarifications are required for the following:

The different processing algorithms of Figure 3 may be briefly described. Figure 3 does not show the third processing algorithm (I think it is normalize_dataset_scale).

In Table 2, what is ˆsk?

Some minor remarks:

  1. It is not difficult to proof that for -> It is not difficult to prove that for (Lines 46, 163)
  2. the require the knowledge -> they require the knowledge (Line 61)
  3. represented with n an the -> represented with n and the (Line 111)

Author Response

Dear Reviewer #1,

Thank you very much for your thorough review of the presented article. You may find the response to your comments inline below. I have used italic font type and gray coloring for quoting your remarks. Following your remarks in the text below, my responses will refer to article lines of its previous version (not the new version, or the version with highlighted changes).

The article ‘On the Optimal Calculation of the Rice Coding’ propose algorithms for the calculation of optimal rice coding parameter. It also validates these algorithms on data coming from multiple sensors, found in the context of real IoT systems. Finally, it also explores the use of multiple rice coding parameter by partitioning the data into subsequences. The article is well written with clear explanations to the different equations and algorithms and adding relevant references to the related works. However, I have some suggestions to the author which I am detailing below.

I am very glad that you found the article readable and understandable in most of its parts.

Though the author has presented the Huffman coding and dictionary methods (Page 3) commonly used for the purpose of data compression as well as their limitations, the suggestion that the Rice coding may be useful for IoT devices (Lines 49-50, 75-77) seems not very convincing to me in the introductory section.

The title of the subsection “Motivation” was replaced for “Applications”. A description of the Rice coding is given providing citations to several papers in the domain of audio, image, video and ECG data compression. Later on, I have provided a lengthier explanation of the concrete application, begin developed in our research group, where Rice coding is being used for compressing data from IoT devices. More details about this IoT application will be unveiled in an upcoming publication, which will cite this article under MDPI review (if published), in the following months.

The authors present the algorithmic complexity of the proposed algorithms, without any explanation on how these values were obtained. Take for example, lines 206-209. Even though, for some of the proposed algorithms, it may seem quite evident, it would still be interesting to have a brief description on how these values were obtained.

Thanks for pointing out the fact that the algorithms complexity was derived without a careful explanation. Lines 206-209 (related to Algorithm 1), lines 340-341 (related to Algorithm 2), Lines 361-363 (related to Algorithm 3), lines 400-406 (related to Algorithm 4) were modified, in order to provide a better understanding of the mentioned complexities. Lines numbering for all algorithms (not only for Algorithm 1) are now visible.

In the context of using single rice coding parameter, the decoding algorithm is presented (Lines 28-34). However, the decoding in the case of the proposed multi-parameter rice coding is not detailed. How these multiple parameters are handled or transmitted? In case of single rice coding parameter, these may be part of the first few bytes or already by the transmitter and the receiving end. But what happens when you have multiple parameters. There is multiple information like the total number of parameters and the parameter values. It becomes more complicated, especially when we see that sub-sequences may be of varying length (Table 4). What is the size overhead (in bytes) to store/transmit this information? Are these considered in the evaluation of the algorithms/heuristics?

I acknowledge that this was not fully described in the article in its first version. A new subsection (Subsection 5.5) is included in order to clarify this aspect. The calculations made in both result sections (Sections 4 and 6) consider this overhead already.

Some clarifications are required for the following: The different processing algorithms of Figure 3 may be briefly described. Figure 3 does not show the third processing algorithm (I think it is normalize_dataset_scale).

I apologize for omitting this in the first version of the article. Lines 232-244 of the article did provide an explanation of the three pre-calculation methods. However, I did forget providing a short name for each pre-calculation method for further references in the article (and hence in Figure 3). Those names are now included in lines 232-244. Figure 3 itself was slightly modified in order to avoid cutting the labels of the bar groups below the figure.

In Table 2, what is ˆsk?

The value ^s_k was explained in lines 326-333. However, it was not included in the caption of Table 2. I apologize for omitting this in the first version. The caption is now extended with an explanation of this value.

Some minor remarks:

  1. It is not difficult to proof that for -> It is not difficult to prove that for (Lines 46, 163)
  2. the require the knowledge -> they require the knowledge (Line 61)
  3. represented with n an the -> represented with n and the (Line 111)

All these typos are now corrected. Thanks for pointing them out.

Other corrections from the author:

  • Algorithm 3 was minimally corrected in three ways: 1) The variable N` was initialized, 2) the second inner loop between lines 14-19 was merged (after having corrected the bounds of this second inner loop) with the first inner loop between lines 9-11, for better readability, and 3) the variable S was removed. These changes do not affect the output results of this algorithm nor its complexity.

Sincerely,

Fernando Solano

Reviewer 2 Report

The article "On the Optimal Calculation of the Rice Coding Parameter" addresses a relevant topic. The results are scientifically sound and they are presented in an understandable manner.

Besides the discrete optimization of partitioning and partition sizes, how does your approach compare to statistics based approaches, such as "Adaptive run-length/Golomb-Rice encoding of quantized generalized Gaussian sources with unknown statistics" by H. Malvar (DOI: 10.1109/DCC.2006.5). A benchmark / comparison may be too laborious, but I suggest to include an educated guess in the related work section.

Additionally, a few minor remarks / typos:

  • in line 61: ... ratio, the_y_ require ...
  • in line 118 & 119: please add _weighted_ sum mentioning the weights which are the probabilities of a symbol being in a sequence
  • in line 124: "for some positive value of p" can be specified to "for some probability p \in (0, 1)"
  • in line 132: the constant value phi could be called by its name - the golden ratio
  • in line 185: the expression in Eq. 22 is just the binary representation of a; stating such facts may help understanding the formulas
  • in Algorithm 1: please insert empty lines for readability; e.g. between 12/13, 16/17
  • in Algorithm 1: L() is used but not defined -- please add a comment referencing Eq. (10)
  • in Figure 3: in my PDF viewer the legend is cut-off, the legend of the blue color is cut at "normalize_dataset_s"
  • in Algorithm 2: as in Algorithm 1, please ref. Eq. 10
  • in Figure 13: the scale of the legend (the percentage values "84% ... ) is overlapping with itself; the values are hardly readable

Author Response

Dear Reviewer #2,

Thank you very much for your thorough review of the presented article. You may find the response to your comments inline below. I have used italic font type and gray coloring for quoting your remarks. Following your remarks in the text below, my responses will refer to article lines of its previous version (not the new version, or the version with highlighted changes).

The article "On the Optimal Calculation of the Rice Coding Parameter" addresses a relevant topic. The results are scientifically sound and they are presented in an understandable manner.

I am very glad that you found the article readable and understandable in most of its parts.

Besides the discrete optimization of partitioning and partition sizes, how does your approach compare to statistics based approaches, such as "Adaptive runlength/Golomb-Rice encoding of quantized generalized Gaussian sources with unknown statistics" by H. Malvar (DOI: 10.1109/DCC.2006.5). A benchmark / comparison may be too laborious, but I suggest to include an educated guess in the related work section.

Thank you very much for this very interesting reference. This article is now included in the bibliography and referenced in Subsection 2.4 of the submitted article, together with a guess on its performance.

Additionally, a few minor remarks / typos:

  • in line 61: ... ratio, the_y_ require ...
  • in line 118 & 119: please add _weighted_ sum mentioning the weights which are the probabilities of a symbol being in a sequence
  • in line 124: "for some positive value of p" can be specified to "for some probability p \in (0, 1)"
  • in line 132: the constant value phi could be called by its name - the golden ratio
  • in line 185: the expression in Eq. 22 is just the binary representation of a; stating such facts may help understanding the formulas
  • in Algorithm 1: please insert empty lines for readability; e.g. between 12/13, 16/17
  • in Algorithm 1: L() is used but not defined -- please add a comment referencing Eq. (10)
  • in Figure 3: in my PDF viewer the legend is cut-off, the legend of the blue color is cut at "normalize_dataset_s"
  • in Algorithm 2: as in Algorithm 1, please ref. Eq. 10
  • in Figure 13: the scale of the legend (the percentage values "84% ... ) is overlapping with itself; the values are hardly readable 

The text was modified following all recommendations of the reviewer. Thank you for helping making the text clearer and more understandable for future readers.

Sincerely,

Fernando Solano

Round 2

Reviewer 1 Report

First of all, I would like to acknowledge the author for taking into account my review comments.

The author has introduced ‘Applications’ sub-section, detailing their previous use of Rice coding for different use cases related to IoT. Table 2 has been clarified and complexity calculations of the different algorithms has been detailed under the code. Figure 3 has been corrected.

One of my major comment was the missing description of the handling of mutli-parameter rice encoding, which the author rectified by adding a section 5.5, explaining the use of negative zero as delimiter (proposal).

Minor comments

  1. Line 49: Rice coding have -> Rice coding has